# Evaluation of Robust Spatial Pyramid Pooling Based on Convolutional Neural Network for Traffic Sign Recognition System

**Christine Dewi [1,2]**, **Rung-Ching Chen [1],***  and **Shao-Kuo Tai [1],***

[1]  Department of Information Management, Chaoyang University of Technology, Taichung 41349, Taiwan; christine.dewi13@gmail.com
[2]  Faculty of Information Technology, Satya Wacana Christian University, Central Java 50711, Indonesia
*  Correspondence: crching@cyut.edu.tw (R.-C.C.); sgdai@cyut.edu.tw (S.-K.T.)

**Abstract:** Traffic sign recognition (TSR) is a noteworthy issue for real-world applications such as systems for autonomous driving as it has the main role in guiding the driver. This paper focuses on Taiwan's prohibitory sign due to the lack of a database or research system for Taiwan's traffic sign recognition. This paper investigates the state-of-the-art of various object detection systems (Yolo V3, Resnet 50, Densenet, and Tiny Yolo V3) combined with spatial pyramid pooling (SPP). We adopt the concept of SPP to improve the backbone network of Yolo V3, Resnet 50, Densenet, and Tiny Yolo V3 for building feature extraction. Furthermore, we use a spatial pyramid pooling to study multi-scale object features thoroughly. The observation and evaluation of certain models include vital metrics measurements, such as the mean average precision (mAP), workspace size, detection time, intersection over union (IoU), and the number of billion floating-point operations (BFLOPS). Our findings show that Yolo V3 SPP strikes the best total BFLOPS (65.69), and mAP (98.88%). Besides, the highest average accuracy is Yolo V3 SPP at 99%, followed by Densenet SPP at 87%, Resnet 50 SPP at 70%, and Tiny Yolo V3 SPP at 50%. Hence, SPP can improve the performance of all models in the experiment.

**Keywords:** spatial pyramid pooling; Yolo V3; object recognition; convolutional neural network

## 1. Introduction

In all countries, traffic signs have essential information for drivers on the road, including the speed limitation, direction indication, stop information, and so on [1]. Traffic sign recognition systems (TSRS) are crucial in numerous applications in the real world, such as autonomous driving, traffic surveillance, driver protection and assistance, road network sustenance, and investigation of traffic disturbances [1]. Two related subjects that important in TSRS are traffic sign detection (TSD) and traffic sign recognition (TSR). TSD directly affects the safety of drivers and because of their ignorance can easily cause damage. Automatic systems that support drivers can improve unsafe driving behavior based on the detection and recognition of signs [2]. TSRS are difficult and complicated tasks in consequence of several problems, including occlusion, illumination, color variation, rotation, and skew that appear from camera setup in the surroundings. Further, there could be multiple signs in an image with different colors, sizes, and shapes [3,4].

The reality of traffic signs deliberates to have distinguishable features and to be specific, such as simple shapes and uniform colors. The detection and recognition of traffic signs imply a constrained problem. In addition, there are some differences in the design of signs between the countries. In certain cases, there can be significant differences in the design of signs in various countries. These differences are

easier to identify by humans, but for an automated detection system can be a major challenge. Therefore, the development of a strong TSRS system is an important and challenging issue in consequence of the latency in the testing time. For this work, we will focus on Taiwan's prohibitory sign detection and recognition. Our motivation is the lack of a database or research system for Taiwan's traffic sign recognition.

This paper summarizes and examines eight convolutional neural network (CNN) models combined with spatial pyramid pooling (SPP) for object detection. We fine-tune them on Taiwan's prohibitory sign dataset, which is built by ourselves to perform traffic sign detection. Based on our inspection, no other scientific paper investigates diverse deep-learning object detectors specifically tailored to the area of traffic sign detection problems while estimating various key factors, such as mean average precision (mAP), intersection over union (IoU), and time of detection.

The main objectives of this study are as follows: (1) Presentation of a brief survey of object detection algorithms based on CNN, specifically Yolo V3, Yolo V3 SPP, Densenet, Densenet SPP, Resnet 50, Resnet 50 SPP, Tiny Yolo V3, and Tiny Yolo V3 SPP. (2) Examination and evaluation of diverse state-of-the-art object detectors, mainly for the traffic sign detection problem. The performance evaluation of these models includes crucial metrics, such as the mAP, detection time, intersection over union (IoU), and the number of billion floating-point operations (BFLOPS). (3) For more complete and accurate learning of multi-scale object features, we use spatial pyramid pooling to collect different scale local features in the same convolutional layer. (4) The experiments show that Yolo V3 SPP strikes the best accuracy and SPP, which can improve the performance of all models.

The rest of the paper is organized as follows. The materials and methods were discussed in Section 2. Section 3 briefly describes our experiment results. Further, Section 4 analyzes, compares and discusses traffic sign detection results. Subsequently, we give conclusions and future research work in Section 5.

## 2. Materials and Methods

### 2.1. CNN for Object Detection

There have been several classic object recognition networks in the last few years [5], for instance AlexNet [6] (2012), VGG [7] (2014), GoogLeNet [8] (2015–2016), ResNet [9,10] (2016), SqueezeNet [11] (2016), Xception [12] (2016), MobileNet [13] (2017–2018), ShufficNet [14] (2017–2018), SE-Net [15] (2017), DenseNet [16] (2017), and CondenseNet [17] (2017), Initially, the convolutional neural network was developed and enlarged to achieve greater precision accuracy. However, networks have grown smaller and more efficient in recent years. In highly accurate target sensing tasks, the new deep learning algorithms, especially those that apply to CNN, such as You Only Look Once, (Yolo) v3, show huge potential [18]. The multiscale and sliding window approach that produces bounding boxes and scores via CNN can be implemented efficiently within a ConvNet [19], and R-CNN [20]. Besides, R-CNN is also expensive in time and memory, as it executes a CNN forward-pass for all object proposal without sharing computation. To solve this problem, spatial pyramid pooling networks (SPPnets) [21] were introduced to increase the efficiency of R-CNN through computational sharing. SPPnet calculates feature maps from the entire input image only once and then supplies feature in arbitrary-size sub-images to generate fixed-length representations and for detectors training. However, SSPnet eliminates the replicated evaluation of convolutional feature maps, it still needs training in a multi-stage pipeline as the fixed-length feature vectors generated by numerous SPP layers are also moved on to fully-connected layers. Therefore, the whole process is still slow. Certain techniques, including single shot multiBox detector (SSD) [22] and Yolo [23], exemplify all the processing in a single fully-convolutional neural network rather than making a persistent pipeline of regional proposals and object classification. This knowledge conducts to a significantly more expeditious object detector. The one-stage method relies on the end-to-end regression approach technology. Yolo V3 [24] applied Darknet-53 to substitute Darknet-19 as the backbone network and employed multiscale prediction [25].

## 2.2. Spatial Pyramid Pooling (SPP)

In terms of object recognition tasks, spatial pyramid pooling (SPP) [26,27] was significantly victorious. Consider its severity, it is competing among methods that use more complicated spatial models. For the interpretation of the spatial pyramid, the image is split into a range of finer grids at each level of the pyramid. In addition, it is commonly-known as spatial pyramid matching (SPM) [28], a development of the bag-of-words (BoW) model [29], which is one of the most famous and successful methods in computer vision methods. SPP has continued been an important component and superior system to win the competition in the classification [30,31] and detection [32] before the recent ascendance of CNN.

Some benefits of SPP [21] could be explained as follow: First, SPP can produce a fixed-length output despite the input dimension. Second, SPP applies multi-level spatial bins, while the sliding window pooling employs just a single-window size. Next, SPP allows us not only to generate images from arbitrarily sized images for testing but also to feed images with different sizes and scales during training. Additionally, training with variable-size images raises invariance in size and decreases overfitting. In addition, SPP is extremely effective in object detection. In the foremost object detection method R-CNN, the features from candidate windows are obtained through deep convolutional networks. Furthermore, SPP can combine features derived at variable scales to the flexibility of input scales. CNN layers receive some despotic input sizes, but they generate outputs of variable sizes. The softmax classifiers or fully-connected layers require fixed-length vectors. Such vectors can be generated by the BoW approach [29] that pools the features together at the same time. SPP improves the performance of BoW in that stage, and it can preserve spatial information by pooling in local spatial bins. The space bins have proportional sizes to the image size, and regardless of the image size the number of bins is fixed. On the contrary, the sliding window pooling of former deep networks and the number of sliding windows depends on the scale of the data. Hence, to implement the deep network for images of arbitrary sizes, the last pooling layer will substitute with an SPP layer. In the particular spatial bin, we pool the replies of each filter and apply max pooling. The outputs of the spatial pyramid pooling are kM dimensional vectors with the number of bins indicated as M. Further, k is the number of filters in the latest convolutional layer. The fixed-dimensional vectors are the input to the fully-connected layer. By using SPP, the input image can vary in size, which allows not only arbitrary aspect ratios but also enables absolute scales. The input image can resize to any scale and adopt an identical deep network. When the input image is at diverse scales, the network with the equivalent filter sizes will extract features at various sizes and scales. A network structure with an SPP layer can be seen in Figure 1. In our work, the SPP blocks layer is inserted to the Yolo V3, Resnet 50, Densenet, and Tiny Yolo V3 configuration file. Moreover, we use the same SPP blocks layer in the configuration file with a spatial model. The spatial model uses down sampling in convolutional layers to receive the important features in the max-pooling layers. It applies three different sizes of the max pool for each image by using [route]. Different layers −2, −4 and −1, −3, −5, −6 in $conv_5$ were uses in each [route].

## 2.3. Object Detection Architecture

The principal features of each architecture (Yolo V3, Densenet, Resnet 50, Tiny Yolo V3) are summarized in this section.

### 2.3.1. Yolo V3 and Tiny Yolo V3

Yolo V3 was proposed by [24] in 2018. It splits the input image into $(S \times S)$ grids cells [33] with the same size and forecast bounding boxes and probabilities for each grid cell. Yolo V3 uses multi-scale fusion to make predictions and uses a singular neural network to process the complete image. The dimension clusters are applied as prior boxes to predict boundary boxes. Therefore, the k-means method is adopted to carry out dimensional clustering on the target boxes in the dataset and get nine prior boxes of various sizes, which are evenly spread to feature graphs of various scales. Further, Yolo V3 allows individual bounding box anchor for each ground truth object [34]. If the

core point of the object's ground truth drops inside a specific grid, and the grid is responsible for recognizing the object. Figure 2 describes the bounding boxes with the prior dimension and location prediction. As shown in Figure 2, $b_x$, $b_y$, $b_w$, $b_h$ are the $x$, $y$ center coordinates of the width, and height of our prediction. $t_x$, $t_y$, $t_w$, and $t_k$ are the network outputs. Next, $c_x$ and $c_y$ are the top-left coordinates of the grid, whereas $p_w$ and $p_h$ are anchors dimensions for the box [23,35].

$$b_x = \sigma(t_x) + c_x \tag{1}$$

$$b_y = \sigma(t_y) + c_y \tag{2}$$

$$b_w = p_w e^{t_w} \tag{3}$$

$$b_h = p_h e^{t_h} \tag{4}$$

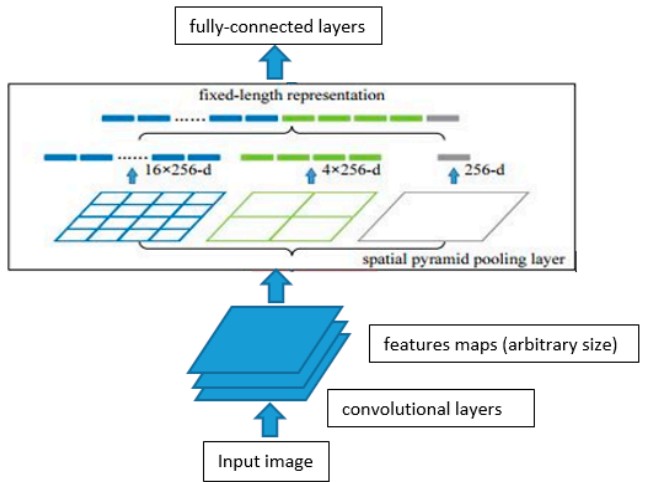

**Figure 1.** A network structure with a spatial pyramid pooling (SPP) layer.

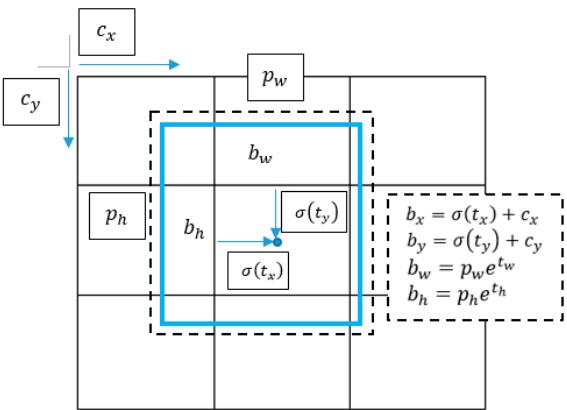

**Figure 2.** Bounding boxes with dimension priors and location prediction.

The Tiny Yolo V3 model is a reduced version of the Yolo V3 model. Yolo V3 applies the architecture of darknet 53 and then employs many $1 \times 1$ and $3 \times 3$ convolution kernels to extract features. This model is lighter and faster than Yolo while also outperforming other light model's accuracy. Tiny Yolo V3 shrinkage the number of convolutional layers, usually it only has seven convolutional layers. The features are derived by a small number of $1 \times 1$ and $3 \times 3$ convolutional layers. In addition, Tiny Yolo V3 uses the pooling layer in place of Yolo V3's convolutional layer with a step size of 2 to attain dimensionality reduction. Nevertheless, its convolutional layer structure still uses the equal structure of the loss function (Convolution2D + BatchNormalization + LeakyRelu) as Yolo V3. The model is trained

and calculated the loss value, and the loss function used by Tiny Yolo V3 is the same as that of Yolo V3. Hence, the loss function is essentially consist of the position of the prediction frame (x,y), the prediction frame size (w,h), the class prediction (class), and the confidence prediction (confidence) [36]. Further, Yolo V3 SPP and Tiny Yolo V3 SPP is implemented by incorporating three SPP modules in Yolo V3 and Tiny Yolo V3 in front of three detection headers between the 5 and 6 convolutional layers [37]. Yolo V3 SPP and Tiny Yolo V3 SPP are designed to improve the detection accuracy of baseline models further.

### 2.3.2. Densenet

Densenet has over 40 layers and has a higher convergence speed [38]. Further, Densenet needs to consider additional functionality channels, including single-level dimensions or cross-level dimensions, to reduce the need for functional replication in the network model and enhance the retrieval of features [39]. Moreover, Densenet has appealing benefits as follows: It assists feature reuse and relieves the disappearing gradient problem. Consequently, it also has clear limitations. First, every layer simply combines the feature maps extracted by concatenating the process from previous layers. The operation was done without considering the interdependencies between different channels [40]. Further, the Densenet is principally composed of Dense Block, Transition Layer, and Growth Rate [41]. Dense Block [42]: every Densenet consists of N Dense Blocks. In any Dense Block, there exist $m$ layers where each layer is linked feed-forward to all consecutive layers. If $x_m$ is denoted as the output from the *mth* layer then it is calculated using Equation (5):

$$x_m = H_m([x_1, x_2, \ldots, x_{m-1}]) \tag{5}$$

where $H_m$ is the composite function that operated in this layer and a concatenation function between the individual layers inside it will be processed. The concatenated features are treated through a combination function that composed of BN, *Relu*, and Convolution ($3 \times 3$).

A layer between each dense block to which the spatial dimension of the characteristic's maps known as the transition layer. It is consisting of ($1 \times 1$) convolution layer and ($2 \times 2$) average pooling. Growth Rate: The output from each concatenation function in Equation (5) is a feature map $f$. The size of the *mth* layers is $f(m-1) + f_0$, where $f_0$ is the number of channels of the major input image. To improve the efficiency of the parameter and to monitor the network growth, f is limited to the growth rate G with a small integer value. This variable helps to monitor the amount of information stored in each layer.

### 2.3.3. Resnet 50

Residual Networks (Resnet) [9] are deep convolutional networks where the basic idea is to skip blocks of convolutional layers by using shortcut connections. Further, Resnet is characterized by a very deep network and contains 34 layers to 152 layers [43,44]. This architecture can be seen in Figure 3 and developed by researchers at Microsoft won the ILSVRC 2015 classification task [45]. In the Resnet model, a residual network structure is implemented. The deep CNN model not only avoids the issue of model deterioration by using the residual network structure, but it also achieves better efficiency. The Resnet used skip connections to make convergence more rapid. Even the much deeper layers of Resnet can be trained more quickly than previous ones. This model also used the batch normalization technique to avoid overfitting [46]. Both of these feature extractors are built with four residual blocks: based on the original paper, the first three-layer (namely *conv2_x, conv3_x, and conv4_x*) extract Region Proposal Networks (RPN) features, while the final layer of *conv4_x* is applied for predicting region proposals. Moreover, box classifier features are gained by the last layer of the fourth residual block (*conv5_x*) [47,48].

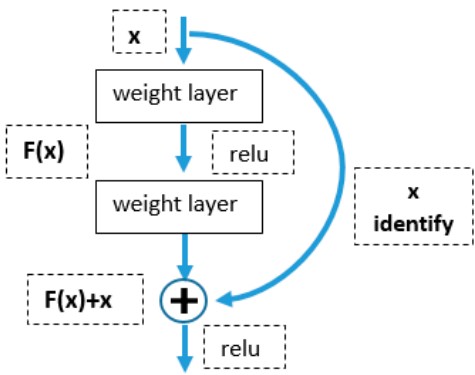

**Figure 3.** Residual block.

*2.4. Methods*

In this section, we explain our proposed methodology to recognize Taiwan's prohibitory signs using spatial pyramid polling combine with Yolo V3. Figure 4 illustrates our Yolo V3 SPP architecture. Algorithm 1 explains the Yolo V3 SPP recognition process as follows.

---

**Algorithm 1** Yolo V3 SPP Recognition Process

---

1.  Split the input image into (S × S) grids.
2.  Create *K* bounding boxes in concert with the estimation of the anchor boxes for every grid.
3.  Extract all object features from the image using the convolutional neural network.
4.  Predict the $\boldsymbol{b} = \begin{bmatrix} b_x, & b_y, b_w, & b_h, & b_c \end{bmatrix}^T$ and the $\boldsymbol{class} = [P1, P2, P3, P4]^T$.
5.  Appeal the optimum confidence $IoU_{pred}^{truth}$ of the *K* bounding boxes with the threshold $IoU_{thres}$.
6.  If $IoU_{pred}^{truth} > IoU_{thres}$ means that the bounding box includes the object. Otherwise, the bounding box does not contain the object.
7.  Select the category with the greatest predicted probability as the object category relating to.
8.  Apply the non-maximum suppression (NMS) to conduct a maximum local search to overcome redundant boxes and output.
9.  Object detection result presentation.

---

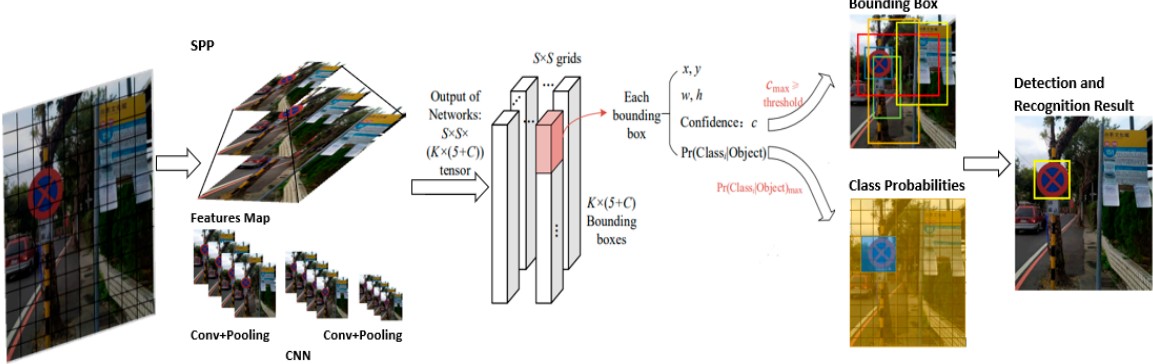

**Figure 4.** Yolo V3 with spatial pyramid pooling (SPP) architecture.

Taiwan's prohibitory sign image class P1, P2, P3, and P4 were used as input of the object detection process. The algorithm processes some phases as follows; (1) the detected targets are limited by bounding boxes. (2) The objects in the class of the image are associated. The same target is given the same mark in each image. (3) The same image will give the same target uniform label. (4) NMS is used to perform a maximum local search to compress redundant boxes and output, then display the results of object detection. In our work, Yolo V3, with a spatial model, uses down sampling in convolutional layers to receive the important features in the max-pooling layers. It applies three different sizes of the

max pool for each image by using [route]. Different layers $-2$, $-4$ and $-1$, $-3$, $-5$, $-6$ in conv$_5$ were uses in each [route].

Yolo V3 SPP model is performed in one phase for detecting and recognizing Taiwan's prohibitory sign. This work used the BBox label tool [49] to generate a bounding box for each sign (no entry, no stopping, no parking, speed limit). Further, the labeling process is executed for all class labels P1, P2, P3, and P4. Usually, one image can have more than one bounding box, and it means one image can have more than one label. In the detection phase, a single class detector model was used, and one class label belongs to one training model. Hence, our experiment uses four training models. Additionally, object coordinates in the form $(x_1, y_1, x_2, y_2)$ are the return values of the bounding box labeling tool. These object coordinates are different from the Yolo input value. On the contrary, the Yolo input value is the center point of the object and its width and height $(x, y, w, h)$. Therefore, the system must modify the bounding box coordinate into the Yolo input format. The modification process use Equations (6)–(11) [50].

$$dw = 1/W \tag{6}$$

$$x = \frac{(x_1 + x_2)}{2} \times dw \tag{7}$$

$$dh = 1/H \tag{8}$$

$$y = \frac{(y_1 + y_2)}{2} \times dh \tag{9}$$

$$w = (x_2 - x_1) \times dw \tag{10}$$

$$h = (y_2 - y_1) \times dh \tag{11}$$

where $H$ is the height of the image, $dh$ is the absolute height of the image, $W$ is the width of the image, and $dw$ is the absolute width of the image. Therefore, float values relative to the width and height of the image $(dw, dh)$; this value can be equal from 0.0 to 1.0.

## 3. Results

### 3.1. Dataset

Considering there is no pre-existing dataset for Taiwan's prohibitory signs, the system had to customize a database and collect the image by ourselves. The dataset split into 70% for training, 30% for testing and the dataset contains pictures of multiple scenes. This experiment focused on Taiwan's prohibitory sign that consists of 235 no entry images, 250 no stopping images, 185 speed limit images, and 230 no parking images. Moreover, Table 1 represents Taiwan's prohibitory signs in detail.

**Table 1.** Taiwan's Prohibitory Signs.

| ID | Class | Name | Sign |
|----|-------|------|------|
| 0 | P1 | No entry | |
| 1 | P2 | No stopping | |
| 2 | P3 | No parking | |
| 3 | P4 | Speed Limit | |

## *3.2. Training Result*

The process of training obtained additional data from the original images with the application of basic geometric transformation methods such as random transformations, rotations, scale shifts, tears, horizontal flips, and vertical flips. These techniques are commonly used to train large neural networks. Therefore, the experiment performs some operations during data augmentation using several parameter settings as follows: rotation_range = 20, zoom_range = 0.10, width_shift_range = 0.2, height_shift_range = 0.2, and shear_range = 0.15. Therefore, the system manually detected and recognized the traffic sign used a bounding box labelling tool to give a coordinate location for the object to be detected [51]. The results of the tools are four points of the position coordinate, along with the class label. Next step, transform the label to Yolo format before training use the Yolo Annotation tool [49]. The tool changes the values to a format that could be read by the Yolo V3 training algorithm. Moreover, the training model environment is a Nvidia RTX1080Ti GPU accelerator 11 GB memory, i7 central processing unit (CPU), and 16 GBDDR2 memory.

The Yolo loss function is as follows [52–54]:

$$loss = \sum_{i-0}^{s^2} coordErr + iouErr + clsErr \tag{12}$$

$$BC(a, \hat{a}) = -[a \log \hat{a} + (1-a) \log(1-\hat{a})] \tag{13}$$

$$ST(w, h) = 2 - w \times h \tag{14}$$

$$iouErr = \sum_{i-0}^{s^2} \sum_{j-0}^{B} I_{ij}^{obj} [BC(c_i, \hat{c}_i)] + \lambda_{noobj} \sum_{i-0}^{s^2} \sum_{j-0}^{B} \{I_{ij}^{noobj} [BC(c_i, \hat{c}_i)] \tag{15}$$

$$clsErr = \sum_{i-0}^{s^2} I_{ij}^{obj} \sum_{c\epsilon classes} BC(p_i(c), \hat{p}_i(c)) \tag{16}$$

$$coordErr = \sum_{i-0}^{s^2} \sum_{j-0}^{B} \{I_{ij}^{obj} \times ST(w_{ij}, h_{ij}) \times [BC(x_i, \hat{x}_i) + BC(y_i, \hat{y}_i)]$$
$$+\lambda_{coord} \sum_{i-0}^{s^2} \sum_{j-0}^{B} \{I_{ij}^{obj} \times ST(w_{ij}, h_{ij}) \times [(w_i, \hat{w}_i)^2 + BC(h_i, \hat{h}_i)]\} \tag{17}$$

where $(\hat{x}, \hat{y}, \hat{w}, \hat{h}, \hat{c}, \hat{p})$ are the central coordinates, width, height, confidence, and category probability of the predicted bounding box, and those symbols without the cusp are real labels. $B$ symbolizes that any grid divines $B$ bounding boxes $I_{ij}^{noobj}$ represents that the object drops within the $j$th bounding box of the $i$th grids. $I_{ij}^{noobj}$ exhibits that there are no targets in the bounding box. Further, IouErr is the IoU error. The grid that includes the object and the grid without an object has different weights. Therefore, λnoobj = 0.5 is added to undermine the impact of a large number of grids without objects on the loss value. The classification error is ClsErr. Cross-entropy is used to calculate losses and works only on the grid with a target. Moreover, Yolo V3 employ the sigmoid function as the activation function for the class prediction. The sigmoid function more effectively finishes the issue when the same target has two labels than the softmax function [55]. Furthermore, the coordinate error is CoordErr. The cross-entropy loss is used for the coordinates in the core point, and the variance loss is applied for the width and height. Our experiment set the λcoord to 0.5, means that the errors of width and height in the calculation are less effective. For a coordinate error, the calculation will be done when the grid predicts an object [53].

Figure 5 explains the reliability of the training process using Yolo V3 (a) and Yolo V3 SPP (b). The training loss value for each model is 0.0141 and 0.0125, respectively. Our work uses max_batches = 8000 iterations, policy = steps, scale = 0.1, 0.1 and steps = 6400, 7200. At the beginning of the training process, the system is beginning with zero or no information and a high learning rate. Therefore, as the neural

network is presented with growing amounts of data, the weights must change less aggressively. Thus, the learning rate needs to be decreased over time. Further, in the configuration file, this decrease in learning rate is accomplished by first specifying that our learning rate decreasing policy is stepwise. For instance, the learning rate starts from 0.001 and remains constant for 6400 iterations. It then multiplies by scales to obtain the new learning rate. If the scale = 0.1, 0.1 and the current iteration number is 1000 (0.001) then current_learning_rate = learning_rate × scales [0] × scales [1] = 0.001 × 0.1 × 0.1 = 0.00001. From Figure 5, we can conclude that Yolo V3 SPP is more stable than Yolo V3 during the training process.

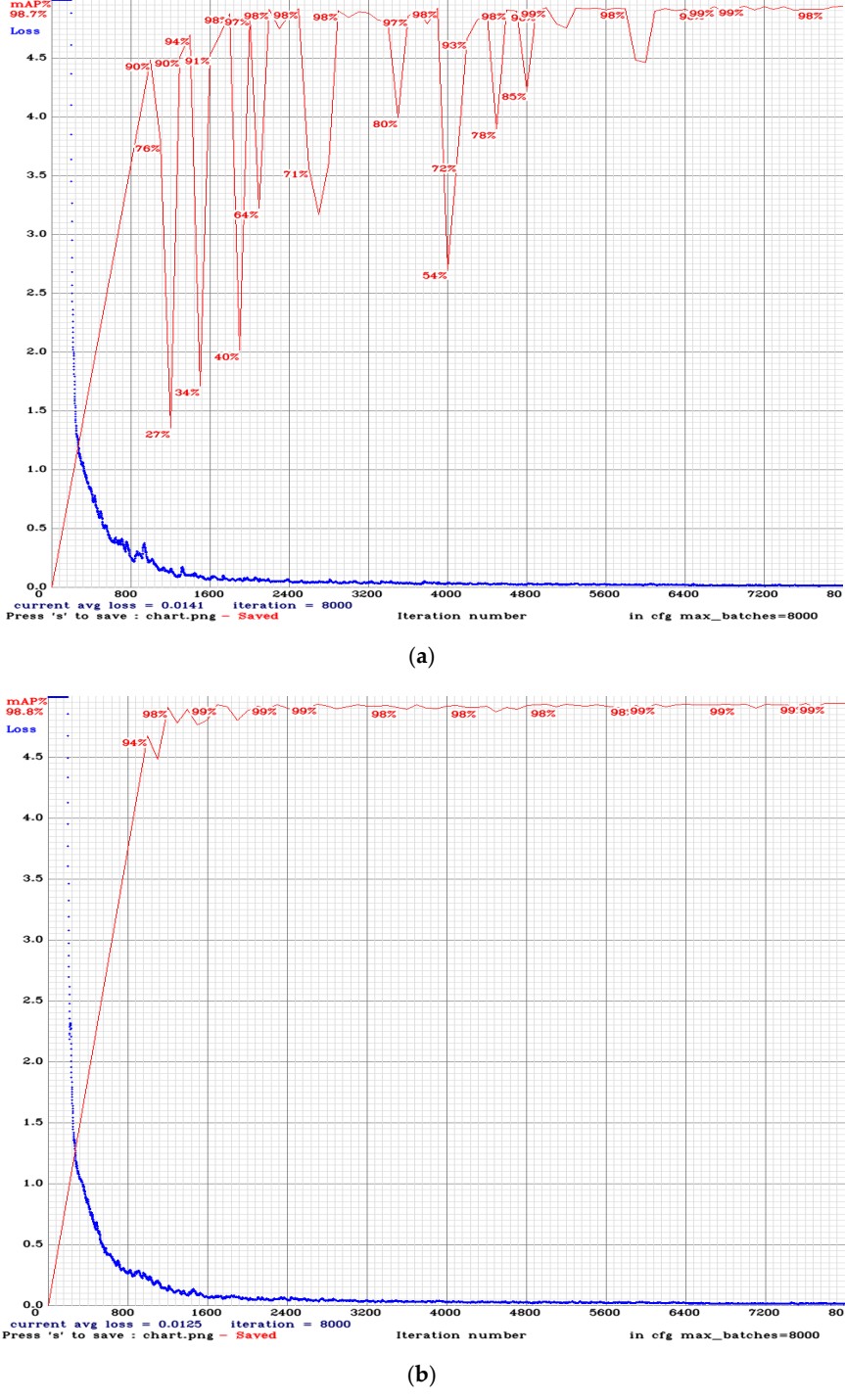

**Figure 5.** Training loss value, mean average precision (mAP), and average precision (AP) performance for all classes using Yolo V3 (**a**) and Yolo V3 SPP (**b**).

Figure 6a describes the dependability of the training process using Densenet. The training process remains stable after 4500 epochs and finishes at 40,000 iterations. During training, Densenet uses max_batches = 45,000, mask_scale = 1, and the training loss value reach 0.0031. In the Figure 6b, Densenet SPP uses max_batches = 45,000, mask_scale = 1, and the iteration remain constant at 9800 epochs with the loss value at 0.0078.

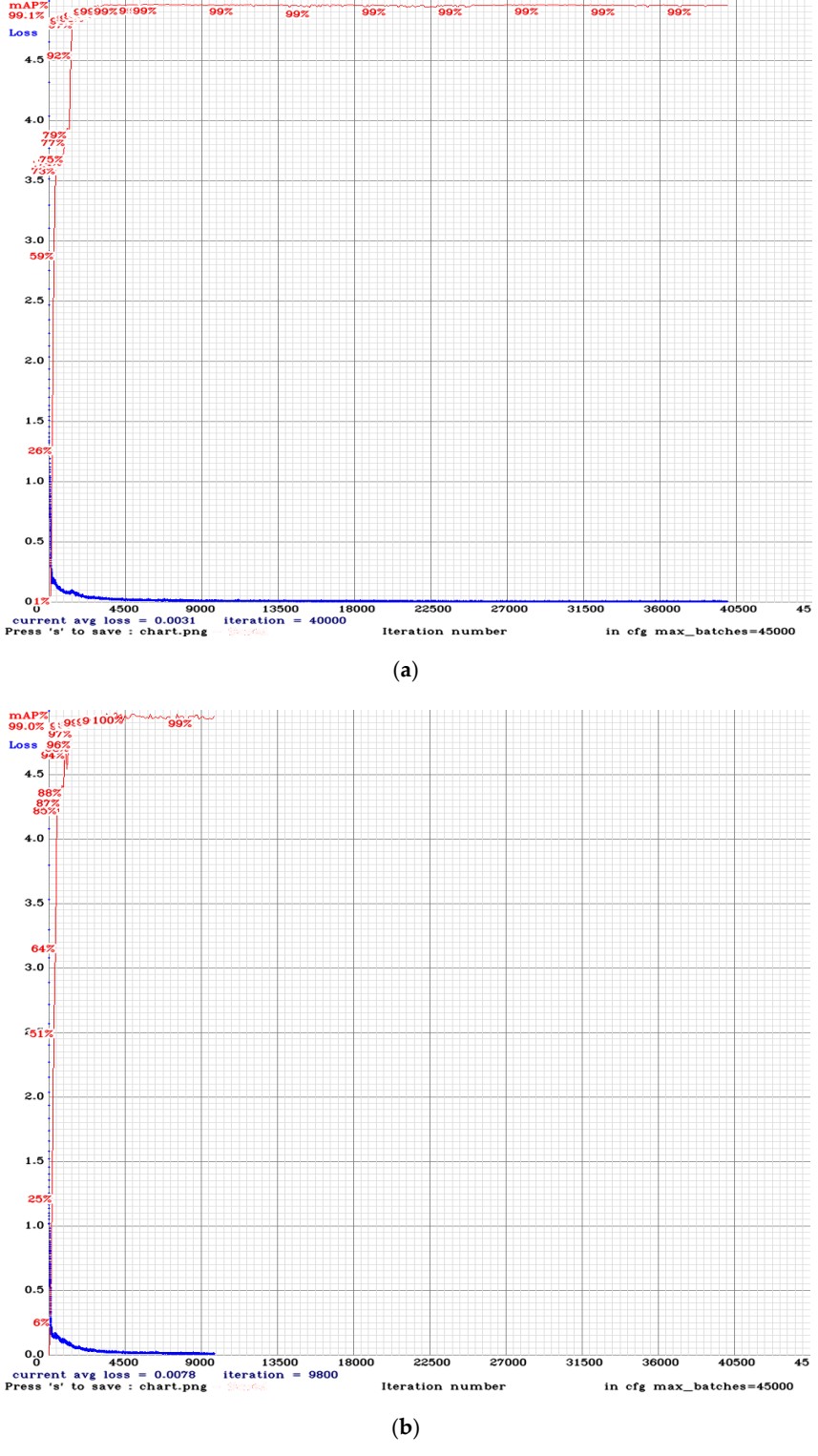

(**a**)

(**b**)

**Figure 6.** Training loss value, mAP, and AP performance for all classes using Densenet (**a**) and Densenet SPP (**b**).

Figure 7a shows the persistence of the training process using Resnet 50. The training stage remains stable after 4500 epochs. Resnet 50 uses max_batches = 45,000, mask_scale = 1, and the training loss value reach 0.0040. Further, Resnet 50 SPP uses max_batches = 45,000, mask_scale = 1, and the iteration remain constant at 28,000 epochs with the loss value at 0.0045 in the Figure 7b.

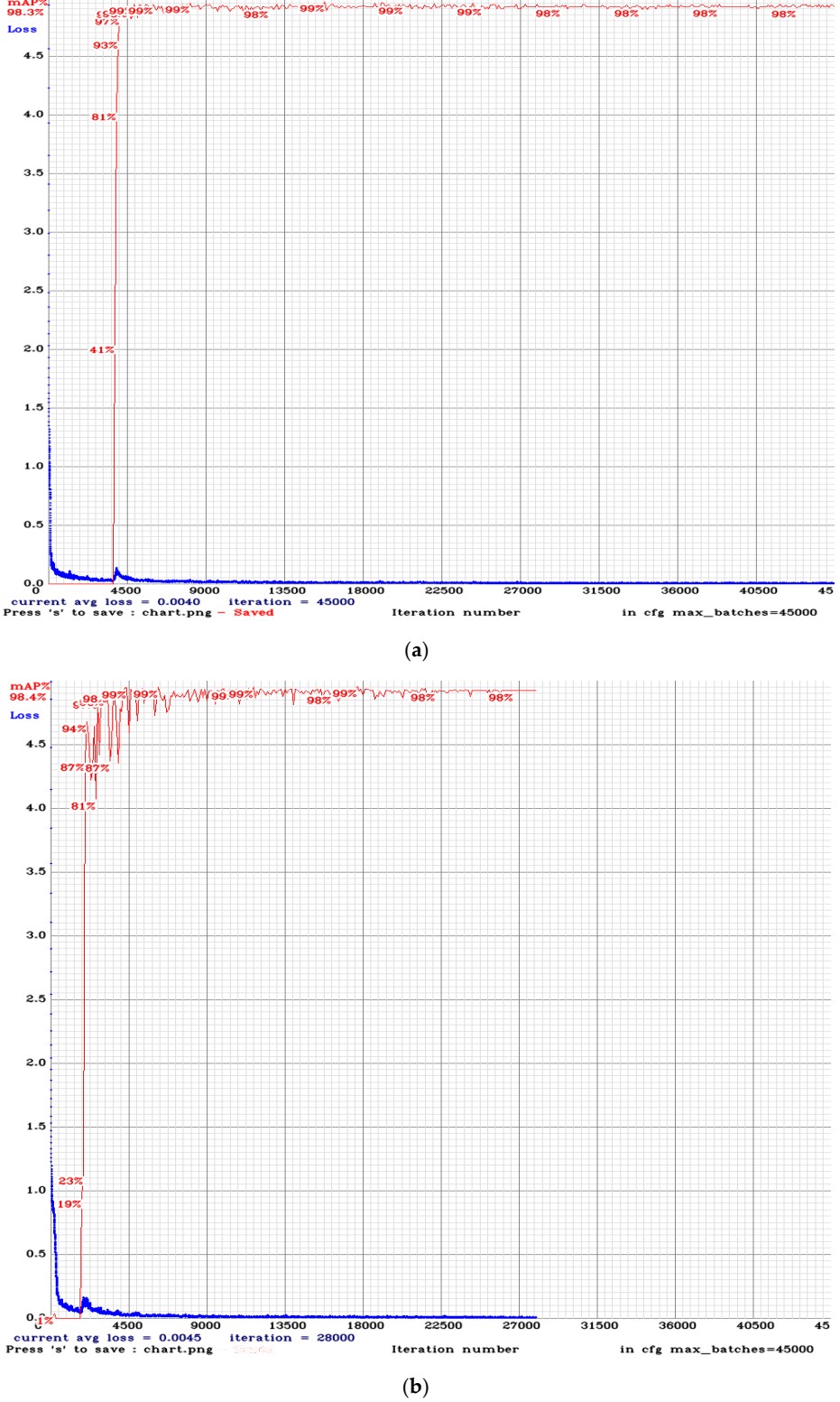

**Figure 7.** Training loss value, mAP, and AP performance for all classes using Resnet 50 (**a**) and Resnet 50 SPP (**b**).

Training loss value, mAP, and AP performance for all classes using Tiny Yolo V3 could be seen in Figure 8. Figure 8a shows the reliability of the training process using Tiny Yolo V3. Tiny Yolo V3 uses max_batches = 500,200, and the training loss value reach 0.0185 at 84,300 iterations. Therefore, Tiny Yolo V3 SPP uses max_batches = 500,200, and the iteration stops at 72,700, with the loss value 0.0144 in Figure 8b. The training process is unstable, and it takes a long time to train this model. The complete results of training mAP and *AP* performance of all models and classes are shown in Table 2.

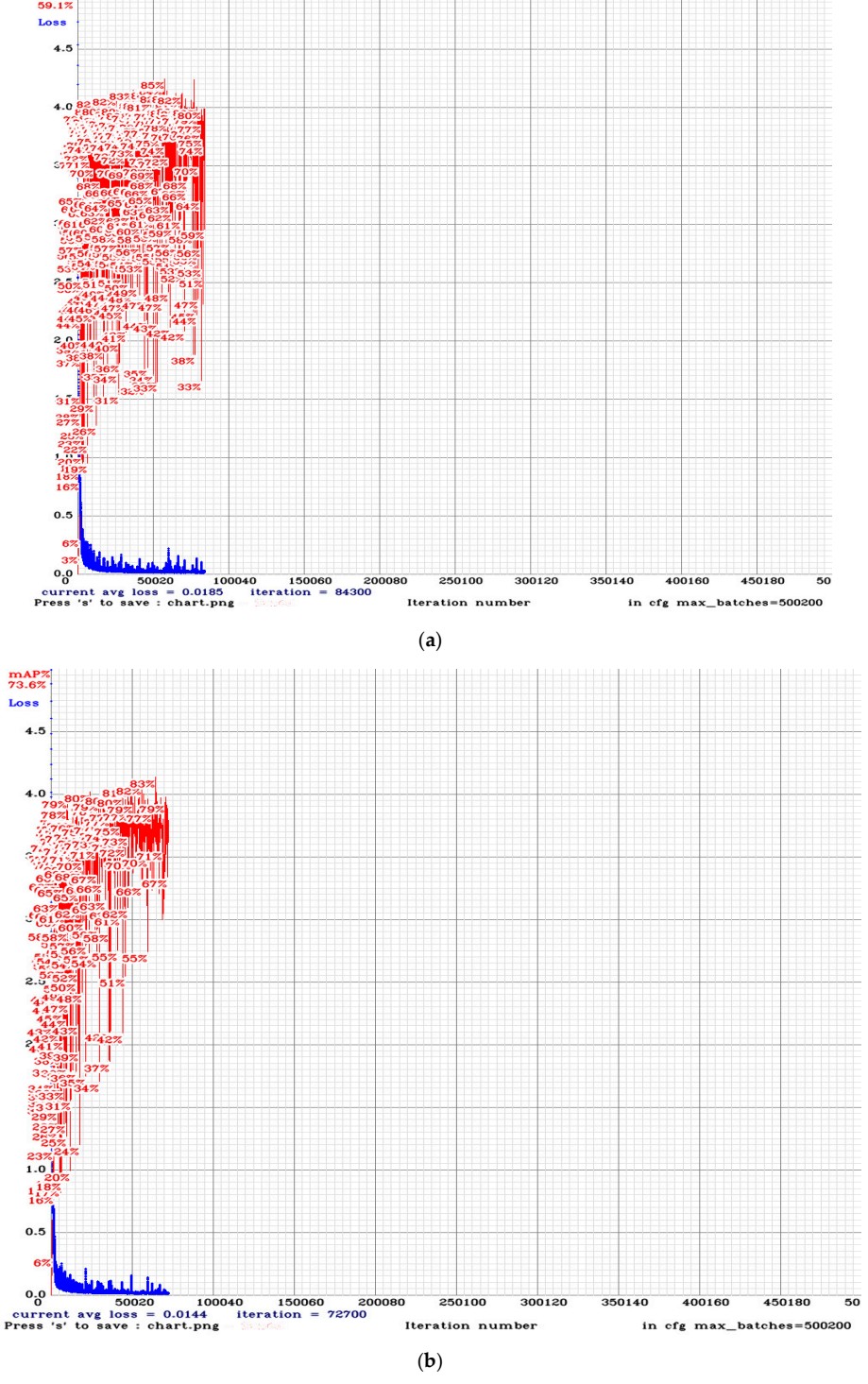

(**a**)

(**b**)

**Figure 8.** Training loss value, mAP, and AP performance for all classes using Tiny Yolo V3 (**a**) and Tiny Yolo V3 SPP (**b**).

**Table 2.** Training loss value, mAP, and AP performance for class P1, P2, P3, and P4.

| Model | Loss Value | Name | AP (%) | TP | FP | Precision | Recall | F1-Score | *IoU* (%) | *mAP@0.50* (%) |
|---|---|---|---|---|---|---|---|---|---|---|
| Yolo V3 | 0.0141 | P1<br>P2<br>P3<br>P4 | 97.5<br>98.8<br>99.9<br>98.7 | 77<br>83<br>62<br>76 | 0<br>0<br>1<br>2 | 0.99 | 0.99 | 0.99 | 88.20 | 98.73 |
| Yolo V3 SPP | 0.0125 | P1<br>P2<br>P3<br>P4 | 97.5<br>98.8<br>99.9<br>98.9 | 78<br>83<br>62<br>79 | 0<br>0<br>1<br>3 | 0.99 | 0.99 | 0.99 | 90.09 | 98.88 |
| Densenet | 0.0031 | P1<br>P2<br>P3<br>P4 | 97.4<br>100<br>100<br>99.9 | 78<br>83<br>62<br>76 | 2<br>0<br>2<br>2 | 0.98 | 0.99 | 0.99 | 88.19 | 98.33 |
| Densenet SPP | 0.0078 | P1<br>P2<br>P3<br>P4 | 98.8<br>100<br>100<br>99.4 | 78<br>82<br>62<br>75 | 1<br>1<br>1<br>2 | 0.98 | 0.99 | 0.99 | 88.55 | 98.53 |
| Resnet 50 | 0.004 | P1<br>P2<br>P3<br>P4 | 96.2<br>98.2<br>97.2<br>96 | 75<br>80<br>59<br>74 | 4<br>3<br>7<br>9 | 0.93 | 0.96 | 0.94 | 73.11 | 97.09 |
| Resnet 50 SPP | 0.0045 | P1<br>P2<br>P3<br>P4 | 97.4<br>100<br>96.6<br>96.8 | 78<br>83<br>61<br>73 | 1<br>0<br>2<br>5 | 0.97 | 0.98 | 0.98 | 79.33 | 97.7 |
| Tiny Yolo V3 | 0.0144 | P1<br>P2<br>P3<br>P4 | 79.6<br>86.0<br>88.7<br>76.4 | 49<br>59<br>52<br>51 | 1<br>2<br>11<br>2 | 0.93 | 0.7 | 0.8 | 75.29 | 82.69 |
| Tiny Yolo V3 SPP | 0.0185 | P1<br>P2<br>P3<br>P4 | 84.2<br>90.0<br>95.4<br>69.6 | 59<br>71<br>56<br>52 | 2<br>0<br>2<br>1 | 0.98 | 0.79 | 0.88 | 79.23 | 84.79 |

Table 2 represents the training loss value, mAP, AP, precision, recall, F1, IoU performance, and calculation time for class P1, P2, P3, and P4. The samples are split into three types: true positive (TP) samples, referring to the number of samples that are properly specified, false positive (FP) samples, referring to the number of samples that have not been identified, true negative (TN) referring to the number of samples that have not been recognized.

Precision (P) and recall (R) [56] are represented by [57,58] in Equations (18) and (19).

$$P = \frac{TP}{TP + FP} \tag{18}$$

$$R = \frac{TP}{TP + FN} \tag{19}$$

Another evaluation index, F1 [59–61] is shown as follows.

$$F1 = \frac{2 \times Precision \times Recall}{Precision + Recall} \tag{20}$$

The integral over the precision p(o) is the average mean average precision (mAP) and shown in Equation (21).

$$mAP = \int_0^1 p(0)do \tag{21}$$

where *p(o)* is the precision of the object detection. *IoU* computes the overlap ratio between the boundary box of the prediction (*pred*) and ground-truth (*gt*) [1].

$$IoU = \frac{Area_{pred} \cap Area_{gt}}{Area_{pred} \cup Area_{gt}} \tag{22}$$

Based on Table 2, Yolo V3 SPP obtains the maximum mAP, around 98.88% with IoU 90.09% followed by Yolo V3 at 98.73% with IoU 88.20%, Densenet SPP at 98.53% with IoU 88.55%, Densenet 98.33% with IoU 88.19%, Resnet 50 SPP at 97.7% with IoU 79.33%, Resnet 50 at 97.09% with IoU 73.11%, Tiny Yolo V3 SPP at 84.79% with IoU 79.23%, and Tiny Yolo V3 at 82.69% with IoU 75.29%. The trend is SPP can increase the mAP and IoU on each model in the experiment. SPP can be combined with any model and will strengthen that model. For instance, the worst model in the experiment was Tiny Yolo V3 with mAP 82.69% and IoU 75.29%. In addition, SPP can improve the performance of Tiny Yolo V3, so for Tiny Yolo V3 SPP, the mAP becomes 84.79% (rise 2.1%) and IoU 79.23 (rise 3.94%).

## 4. Discussion

In this stage, we use twenty Taiwan's prohibitory sign images for testing with different sizes and conditions. The accuracy and time measurements of the experiments are presented in Table 3. In general, Yolo V3 SPP exhibits better accuracy than other models. The highest average accuracy is Yolo V3 SPP at 99% followed by Yolo V3 at 92%, Densenet SPP at 87%, Densenet at 82%, Resnet 50 SPP at 70%, Resnet 50 at 50%, Tiny Yolo V3 SPP at 50%, and Tiny Yolo V3 at 40%. The trend is that the accuracy of the combination model with SPP increases with the detection time, which is means that the combination model with SPP involves more time to detect the sign. For instance, for Yolo V3 SPP, the average time of detection is 17.6 milliseconds, while Yolo V3 requires 16.7 millisecond. The longest detection time is Densenet SPP; this model requires around 40 milliseconds. Following this, Densenet needs 38.3 milliseconds to detect the sign. On the other hand, the fastest model in the experiment is Tiny Yolo V3. Therefore, Tiny Yolo V3 needs 5.4 milliseconds, and Tiny Yolo V3 SPP requires 8 milliseconds to recognize the sign. Further, the SPP affects the performance of accuracy and detection time. In the experiment result, the images are tested one by one to show that SPP can improve the detection and recognition performance of traffic signs compared to those not using SPP. For example, there are 5 images that cannot be detected using Resnet 50 models, but Resnet 50 SPP can detect all traffic signs in the images properly as shown in Table 3.

**Table 3.** Testing accuracy using Taiwan's prohibitory sign images of various sizes.

| Image | Yolo V3 | | Yolo V3 SPP | | Densenet | | Densenet SPP | | Resnet 50 | | Resnet 50 SPP | | Tiny Yolo V3 | | Tiny Yolo V3 SPP | |
|---|---|---|---|---|---|---|---|---|---|---|---|---|---|---|---|---|
| | acc | ms | acc | ms | acc | ms | acc | ms | acc | ms | acc | ms | acc | ms | acc | ms |
| 1 | 0.97 | 15 | 0.97 | 14.4 | 0.73 | 48 | 0.81 | 24 | 0.84 | 16.9 | 0.49 | 20.7 | 0.35 | 6.4 | 0.44 | 12.9 |
| 2 | 0.9 | 20 | 0.98 | 21.3 | 0.92 | 27.5 | 0.89 | 51.9 | 0.68 | 15.6 | 0.6 | 22.5 | - | 4.9 | - | 3.2 |
| 3 | 0.83 | 19 | 0.99 | 15.2 | 0.94 | 41.3 | 0.92 | 21.1 | 0.79 | 15.3 | 0.86 | 24.3 | - | 2.8 | - | 5.3 |
| 4 | 0.95 | 21 | 0.99 | 26.1 | 0.8 | 49 | 0.76 | 48.8 | - | 16.7 | 0.61 | 24.1 | 1 | 4.7 | 0.99 | 3.5 |
| 5 | 0.89 | 18 | 0.99 | 14.8 | 0.85 | 45.9 | 0.83 | 21 | 0.91 | 9.2 | 0.91 | 11.8 | - | 2.7 | 0.37 | 3.9 |
| 6 | 0.98 | 18 | 0.99 | 14.4 | 0.87 | 30.9 | 0.86 | 41.1 | - | 15.7 | 0.51 | 10.6 | - | 3.2 | - | 6.3 |
| 7 | 0.96 | 20 | 1 | 23.2 | 0.85 | 46.4 | 0.88 | 47.9 | - | 16.6 | 0.81 | 23 | 1 | 2.8 | 0.79 | 21.3 |
| 8 | 1 | 16 | 1 | 16.8 | 0.9 | 30.2 | 0.96 | 45.9 | 0.6 | 15.8 | 0.9 | 17.6 | 0.99 | 3.1 | 0.9 | 13.9 |
| 9 | 0.88 | 21 | 0.99 | 15.1 | 0.76 | 40.6 | 0.74 | 29.2 | - | 15.5 | 0.64 | 20.3 | - | 14.3 | - | 3.3 |
| 10 | 0.9 | 15 | 0.98 | 32.6 | 0.63 | 28.5 | 0.82 | 53.2 | - | 19.8 | 0.51 | 24.1 | - | 2.7 | 0.94 | 3.3 |
| 11 | 0.96 | 22 | 0.99 | 14.5 | 0.82 | 28.6 | 0.96 | 34.9 | 0.39 | 16.1 | 0.43 | 16.8 | 0.54 | 10.3 | - | 8.5 |
| 12 | 0.79 | 15 | 0.99 | 15 | 0.74 | 27.2 | 0.95 | 42.9 | 0.55 | 9.4 | 0.68 | 10.2 | 0.98 | 4.4 | 0.87 | 3.5 |
| 13 | 1 | 14 | 1 | 16.6 | 0.92 | 45.8 | 0.95 | 49.1 | 0.93 | 13.9 | 0.94 | 10 | - | 3.5 | - | 3.3 |
| 14 | 1 | 15 | 1 | 17.2 | 0.92 | 37.3 | 0.95 | 36.5 | 0.85 | 21.9 | 0.92 | 25.6 | - | 3.8 | - | 7.1 |
| 15 | 0.92 | 15 | 0.97 | 14.7 | 0.8 | 45.6 | 0.95 | 27.4 | 0.76 | 15.5 | 0.61 | 11.8 | 0.99 | 19.8 | 0.87 | 19.2 |
| 16 | 0.87 | 15 | 0.99 | 14.9 | 0.79 | 47 | 0.5 | 46.8 | 0.29 | 10.9 | 0.31 | 20 | - | 3.9 | - | 8.4 |
| 17 | 0.98 | 13 | 1 | 14.6 | 0.74 | 42.2 | 0.91 | 59.1 | 0.92 | 9 | 0.69 | 18.8 | - | 2.8 | - | 19.9 |
| 18 | 0.95 | 15 | 1 | 14.8 | 0.89 | 42.1 | 0.9 | 23.9 | 0.9 | 21.1 | 0.87 | 22.5 | 0.99 | 2.7 | 1 | 3.9 |
| 19 | 0.87 | 15 | 0.99 | 21.2 | 0.72 | 24.8 | 0.82 | 43.1 | 0.28 | 9.9 | 0.78 | 25.8 | 0.94 | 3.4 | 1 | 5.8 |
| 20 | 0.88 | 15 | 0.99 | 15.2 | 0.86 | 38 | 0.94 | 52.2 | 0.68 | 20.6 | 0.9 | 24.7 | 0.98 | 5.6 | 0.98 | 3.4 |
| Average | 0.92 | 17 | 0.99 | 17.6 | 0.82 | 38.3 | 0.87 | 40 | 0.52 | 15.3 | 0.7 | 19.3 | 0.4 | 5.4 | 0.5 | 8 |

Convolution subsampling and max-pooling have dissimilar benefits. Hence, Convolution subsampling can be better reversed, probably in the subsequent upsampling layers. Different layers −2, −4 and −1, −3, −5, −6 in $conv_5$ were uses in each [route]. However, Max pooling acts somewhat to remove some high-frequency noise from the image by selecting only max values from the adjacent regions. By combining both, SPP seems to affect the advantages of both, improving the backbone network of YoloV3, Resnet 50, Densenet, and Tiny Yolo V3.

Figure 9a describes the Class P4 recognition result using Yolo V3. Image P4-5.jpg was predicted in 14.7 milliseconds with 88% accuracy. The bounding box coordinate position as follows: left_x: 91, top_y: 202, width: 239, and height: 321. Using the same image, Yolo V3 SPP in Figure 9b reaches the highest accuracy 99% and needs 15.2 milliseconds for detecting time. The coordinate position is left_x: 75, top_y: 231, width: 277, and height: 283. Next, detection results using Densenet and Densenet SPP explain in Figure 9c,d. They use default mask_scale value 1.00 and got 86% and 94% accuracy, respectively. Figure 9e,g show the unique detection using Resnet 50 and Tiny Yolo V3. They draw two bounding boxes in one detection stage with the correct label (class P4). Resnet 50 predict the Class P4 sign with 34% accuracy (left_x: -51, top_y: 222, width: 549, and height: 295) and 68% accuracy (left_x: 68, top_y: 108, width: 270, and height: 536). Then, Tiny Yolo V3 requires 5.4 millisecond to detect the signs and obtains 98% accuracy (left_x: 35, top_y: 47, width: 342, and height: 645) and 97% accuracy (left_x: 60, top_y: 264, width: 251, and height: 212). From the test result in Figure 9, we can conclude that all models can detect the class P4 well with different bounding box coordinate.

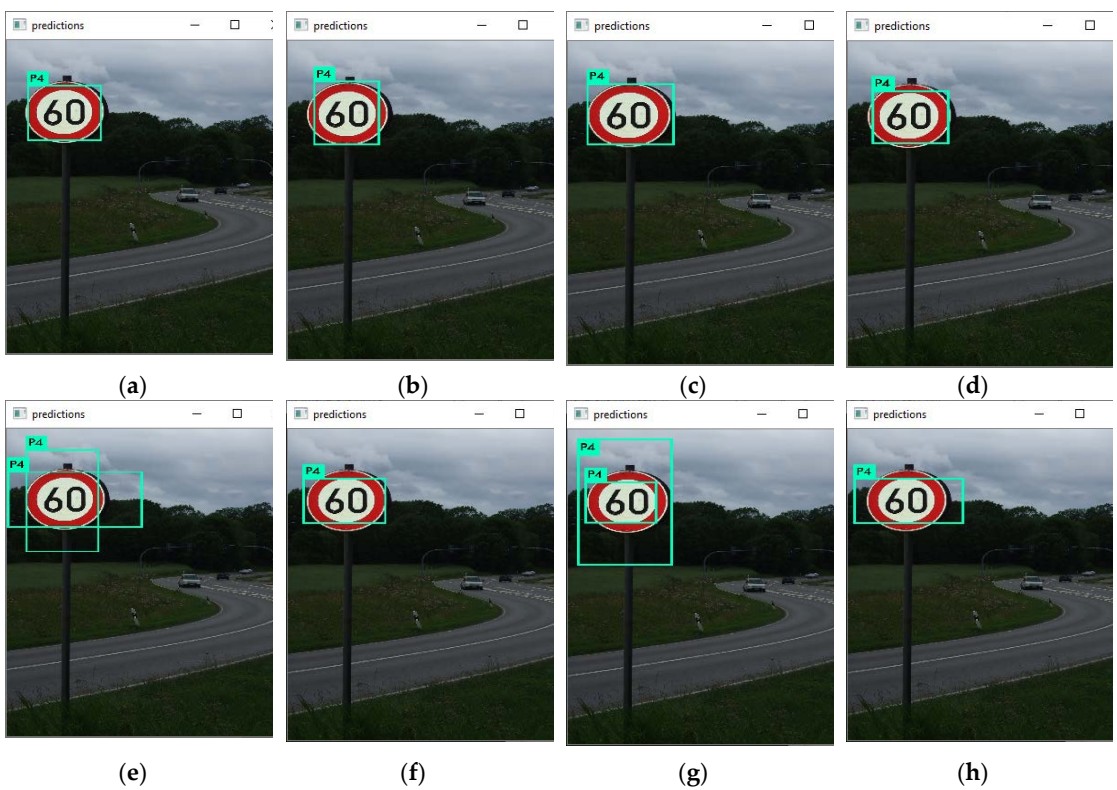

**Figure 9.** Taiwan's prohibitory sign (Class P4, Image P4-5.jpg) recognition result using (**a**) Yolo V3, (**b**) Yolo V3 SPP, (**c**) Densenet, (**d**) Densenet SPP, (**e**) Resnet 50, (**f**) Resnet 50 SPP, (**g**) Tiny Yolo V3, and (**h**) Tiny Yolo V3 SPP.

Figure 10 indicates the comparison of BFLOPS, workspace size, and layers for each model in the experiments. Yolo V3 generates total BFLOPS 65.312, allocate additional workspace size 52.43 MB, and loads 107 layers from weights file. Similar to this, Yolo V3 SPP loads 114 layers and requires workspace size 52.43 with a total BFLOPS of 65.69. Next, Densenet and Densenet SPP load 306 layers and 312 layers, provide a large workspace size 104.86 MB with a total BFLOPS 31.869 and 33.535,

respectively. Furthermore, Resnet 50 and Resnet 50 SPP filled 69 layers and 75 layers, divide additional workspace size 26.33 MB with a total BFLOPS 26.439 and 28.661, successively. Moreover, Tiny Yolo V3 and Tiny Yolo V3 SPP load fewer layers compare to others. They only load 24 layers and 30 layers, requires 52.43 MB workspace size with a total BFLOPS 5.452 and 5.766. In addition, Tiny Yolo V3 loads the fewest layers (24 layers) and BFLOPS (5.452). Densenet SPP contains the most layers (312 layers) and requires a large workspace size (104.86). The highest total BFLOPS achieved by Yolo V3 SPP (65.69). SPP can improve the total BFLOFS 0.378 from 65.312 to 65.69, thus making Yolo V3 SPP more robust, stable, and accurate.

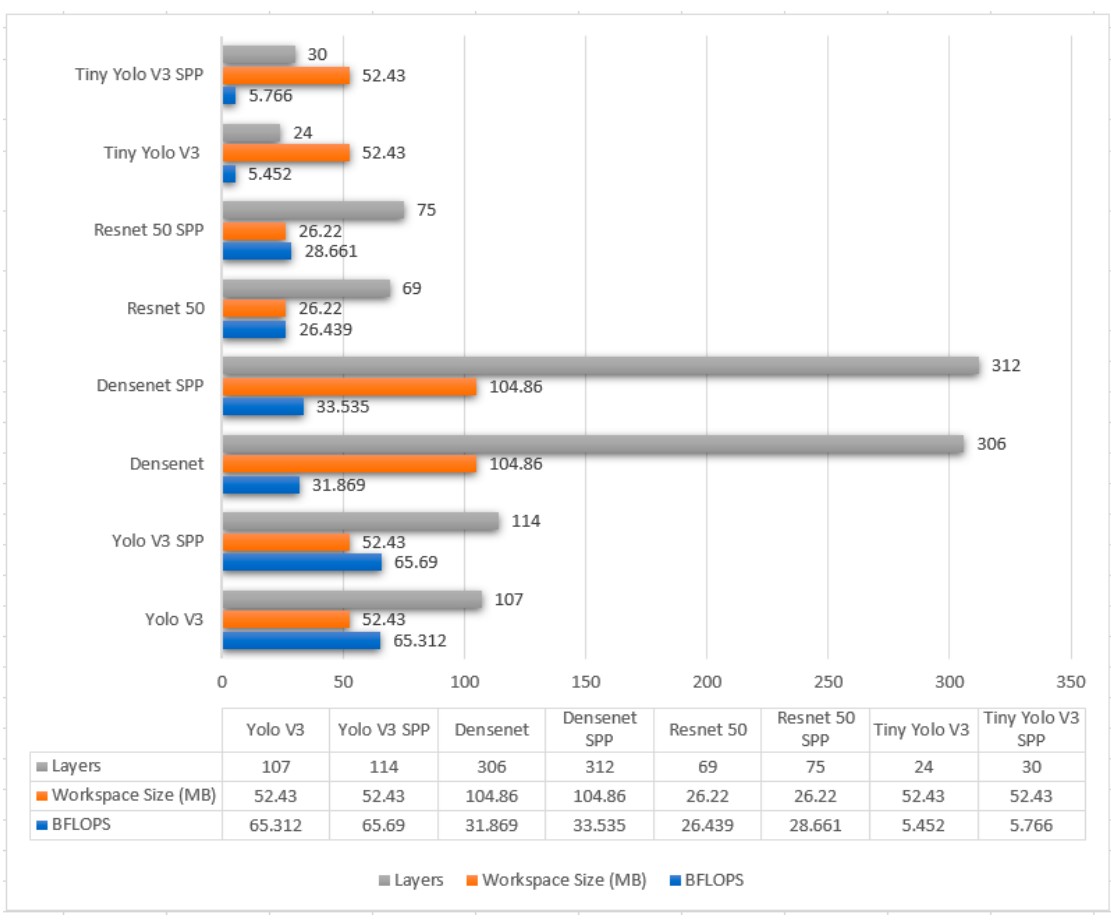

**Figure 10.** Billion floating-point operations (BFLOPS), workspace sizes, and layers comparison.

Figure 11 shows the detection effectiveness of different algorithms. It can be seen that the localization accuracy of Yolo V3 SPP (Figure 11b) was higher than the others. Yolo V3 SPP can detect all two signs in the image. In Figure 11a or Figure 11c–f, all algorithms failed to detect all class P1 signs in the image, detecting only a single sign. However, for the last two images in Figure 11g,h, all algorithms exhibited false detection and missed detection.

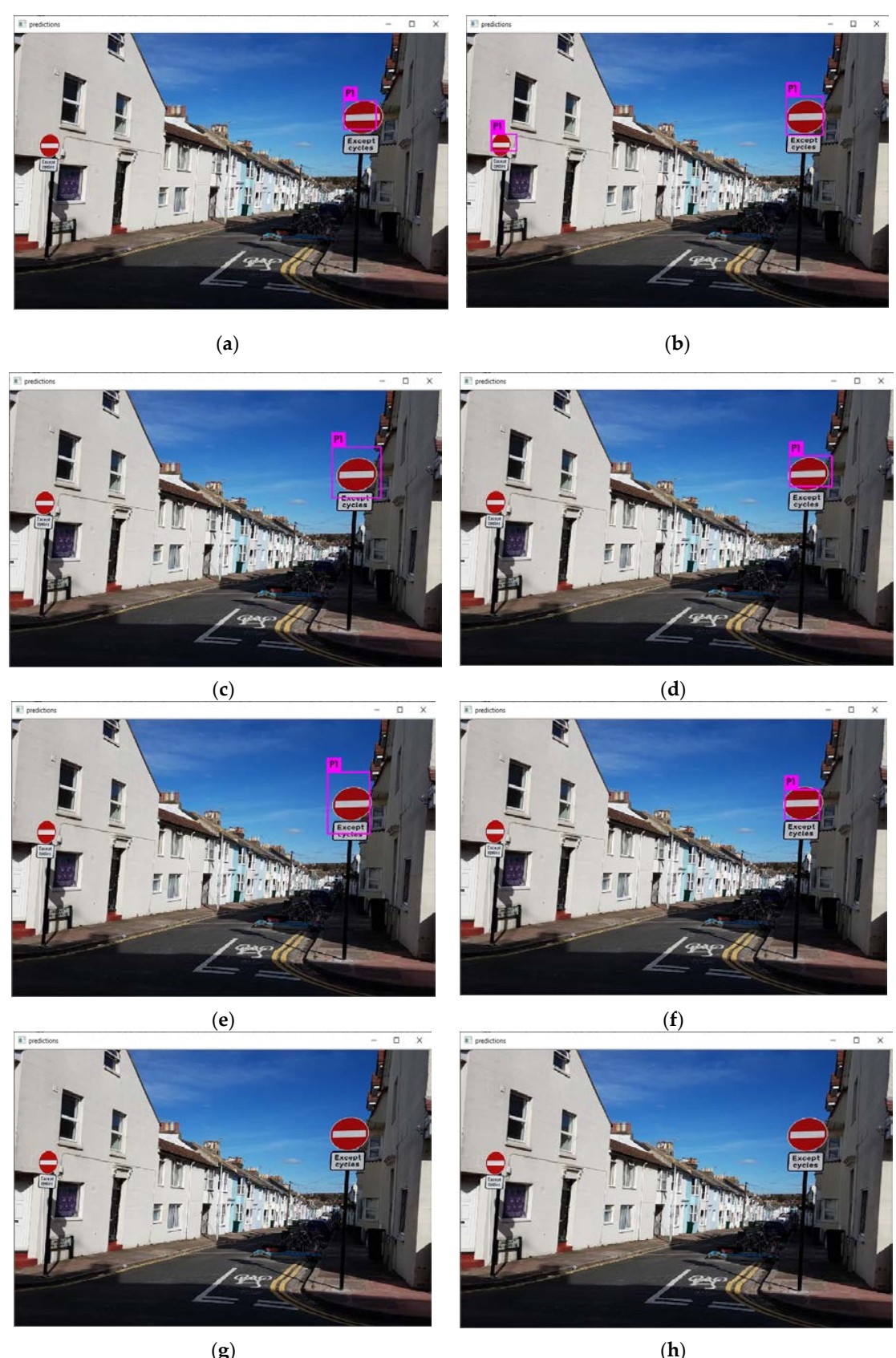

**Figure 11.** Taiwan's prohibitory sign (Class P1) recognition result using (**a**) Yolo V3, (**b**) Yolo V3 SPP, (**c**) Densenet, (**d**) Densenet SPP, (**e**) Resnet 50, (**f**) Resnet 50 SPP, (**g**) Tiny Yolo V3, and (**h**) Tiny Yolo V3 SPP.

## 5. Conclusions

This paper presents an experimental comparative analysis of eight models of traffic signs based on deep neural networks. We investigate the principal aspects of certain detectors, such as precision accuracy, detection time, workspace size, and the number of floating-point operations within CNN. In addition, this paper refers to the spatial pyramid pooling (SPP) and modifies the backbone network of Yolo V3, Resnet 50, Densenet, and Tiny Yolo V3. We employ SPP to raise the local region at diverse scales in the equivalent convolutional layer for learning multi-scale object features more details. The *mAP* comparison of all models shows that Yolo V3 SPP outperforms other models in the experiment. Yolo V3 SPP exhibits the highest total BFLOPS (65.69), and *mAP* (98.88%). SPP can improve the total BFLOFS 0.378 from 65.312 to 65.69, thus making Yolo V3 SPP more robust for detecting the sign. The experimental results disclose that SPP can rectify the effectiveness of detecting and recognizing Taiwan's prohibitory signs. SPP improves the performance and backbone network of YoloV3, Resnet 50, Densenet, and Tiny Yolo V3. Although SPP requires longer time, this model is better for detecting multiple images. As shown in Figure 11b, Yolo V3 SPP can detect all signs in the image while others not. Nevertheless, Tiny Yolo V3 and Tiny Yolo V3 SPP load fewer layers (24 layers) compare to others. Densenet SPP contains the most layers (312 layers) and requires a large workspace size (104.86 MB). Related to the detection time, the fastest models in the experiment are Tiny Yolo V3, and the longest models are Densenet SPP.

In future research work, we will enhance our dataset to all of Taiwan's traffic signs and add experimental data in multiple scenarios and different weathers conditions for training and testing. We will expand the dataset through generative adversarial networks (GAN) [62–64] to obtain better performance and results.

**Author Contributions:** Conceptualization, C.D., R.-C.C. and S.-K.T.; data curation, C.D.; formal analysis, C.D.; funding acquisition, R.-C.C.; investigation, C.D. and R.-C.C.; methodology, C.D. and R.-C.C.; project administration, R.-C.C. and S.-K.T.; software, C.D.; supervision, R.-C.C. and S.-K.T.; validation, R.-C.C.; visualization, C.D.; writing—original draft, C.D.; writing—review and editing, C.D. and R.-C.C. All authors have read and agreed to the published version of the manuscript.

**Funding:** This research was funded by the Ministry of Science and Technology, Taiwan. The Nos are MOST-107-2221-E-324-018-MY2 and MOST-106-2218-E-324-002, Taiwan. This research is also partially sponsored by Chaoyang University of Technology (CYUT) and the Higher Education Sprout Project, Ministry of Education (MOE), Taiwan, under the project name: "The R&D and the cultivation of talent for health-enhancement products."

**Acknowledgments:** The authors would like to acknowledge all the colleagues and partners from Chaoyang University of Technology, Satya Wacana Christian University, and others that take part in this work.

**Conflicts of Interest:** The authors declare no conflict of interest.

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
