# Peer review of "Evaluation of Robust Spatial Pyramid Pooling Based on Convolutional Neural Network for Traffic Sign Recognition System"

_electronics, doi:10.3390/electronics9060889_

Round 1
Reviewer 1 Report
In general comparing the effectiveness of multiple convolutional neural network techniques on a real life scenario such as traffic sign detection has some merit. Also adding multi-resolution techniques to the neural networks adds to the merit.
However, it wasn't clear through the paper starting with the Abstract whether the multi-resolution analysis was added to all the techniques or just one of them. Also, it wasn't clear how the techniques work before and after the added multi-resolution analysis.
Some acronyms were not defined clearly such as IOU: intersection over union. and BFLOps. B stands for billion.
Also, I disagree with the authors that extra BFLOPs is something good. What happened is the multi-resolution technique added to the complexity of the algorithm and increased its convergence time while improving its accuracy.
Some tables and results especially Table 3 are very long and hard to read.
Author Response
Thanks for the comments. We explain the question as follows.
1. In general, comparing the effectiveness of multiple convolutional neural network techniques on a real-life scenario such as traffic sign detection has some merit. Also adding multi-resolution techniques to the neural networks adds to the merit.
However, it wasn't clear through the paper starting with the Abstract whether the multi-resolution analysis was added to all the techniques or just one of them. Also, it wasn't clear how the techniques work before and after the added multi-resolution analysis.
Response:
Thanks for the comments. We use the multi-resolution analysis of all techniques, also for training and testing. We add the explanation inline 16-17 of the abstract. Besides, for the comparison result, we can see in Table 2 (Line 343) and Table 3 (Line 369).
We adopt the concept of SPP to improve the backbone network of Yolo V3, Resnet 50, Densenet, and Tiny Yolo V3 for building feature extraction. Further, we employ spatial pyramid pooling for learning multi-scale object features more comprehensively.
2. Some acronyms were not defined clearly such as IOU: intersection over union. and BFLOPs. B stands for billion.
Response:
We defined the acronyms based on reviewer comment:
- In abstract lines 17-20.
The evaluation and comparison of these models include vital metrics, such as the mean average precision (mAP), workspace size, detection time, intersection over union (IoU), and the number of billion floating-point operations (BFLOPS). Our findings show that Yolo V3 SPP strikes the best total BFLOPS (65.69), and mAP (98.88%).
- Line 55-58.
(2) Analysis and evaluation of several state-of-the-art object detectors tuned, especially for the traffic sign detection task. The evaluation of these models includes critical metrics, such as the mAP, detection time, intersection over union (IoU), and the number of billion floating-point operations (BFLOPS).
3. Also, I disagree with the authors that extra BFLOPs is something good. What happened is the multi-resolution technique added to the complexity of the algorithm and increased its convergence time while improving its accuracy.
Thanks for the comments. In this experiment, we explain that while increasing the accuracy, we also found that the total BFLOPS also increases. The highest mAP of the model also has the highest total BFLOPS.
4. Some tables and results especially Table 3 are very long and hard to read.
Thanks for the comments. We revised Table 3 inline 369.

Reviewer 2 Report
In this paper, the spatial pyramid network is used to improve the deep learning method to detect Taiwan's prohibited traffic signs, and the artificial intelligence technology is combined with practical application. The topic of the thesis is very interesting and the motivation is very beautiful, but the traffic signs are greatly affected by the weather, the scene, the color of the signs, etc., and have many challenges. Once these problems are solved, they will have an inestimable effect on autonomous driving and driverless driving. However, I have some suggestions I hope you can adopt.
P-1, L-13: ”This paper analyses the state-of-the-art of several object-detection systems (Yolo V3, Resnet 50, Densenet, and Tiny Yolo V3) combined with Spatial Pyramid Pooling (SPP).” Resnet 50 and Densene are classification network rather than a target detection network.
P-1, L-19: ” Our findings show that Yolo V3 SPP strikes the best total BFLOPS (65.69), and mAP (98.88%). Hence, SPP can improve the performance of all models in the experiment.” SPP improves the performance of all models, it is recommended to give specific data to illustrate.
P-3, L-113: “ The fixed-dimensional vectors are the input to the fully-connected layer. By using SPP, the input image can vary in size, which allows not only arbitrary aspect ratios but also enables absolute scales.” Different sizes of images, different sizes of images obtained by the same convolution, and then through the full connection layer fixed by neurons are easy to cause problems.
P-3, L-118: “In our work, the SPP blocks layer is inserted to the Yolo V3, Resnet 50, Densenet, and Tiny Yolo V3 configuration file. ” Resnet50 and Densene classification use fully connected layers, while Yolov3 uses convolutional layer prediction. Whether the SPP structure used by three is the same, please explain.
P-3, L-124: “Yolo V3 and Tiny Yolo V3” There are multi-scale predictions for Yolo v3 and yolov3-tiny. In this section, please explain where SPP is added to Yolov3 and yolov3-tiny.
P-7, L-234: “The dataset split into 70% for training and 30% for testing. This experiment focused on Taiwan's prohibitory sign that consists of 235 no entry images,250 no stopping images, 185 speed limit images, and 230 no parking images.” Whether the data set contains pictures of multiple scenes and weather.
P-11, L-332: “ Table 2. Training loss value, mAP, and AP performance for class P1, P2, P3, and P4.” Whether the GPU is used for acceleration in Table 2. If yes, check if the time is wrong, please check carefully.
P-12, L-354: “ Table 3. Testing accuracy using Taiwan’s prohibitory sign images of various sizes.” Testing a single picture is not very persuasive, it is recommended to conduct an overall statistical analysis of the test data set.
It is recommended to add experimental data in multiple scenarios and multiple weathers to show that the algorithm is robust.
Author Response
Thanks for the comments. We explain the question as follows.
- P-1, L-13:” This paper analyses the state-of-the-art of several object-detection systems (Yolo V3, Resnet 50, Densenet, and Tiny Yolo V3) combined with Spatial Pyramid Pooling (SPP).” Resnet 50 and Densenet are classification network rather than a target detection network.
In this research, we use Resnet 50 and Densenet as a classification network or classic networks related to Object Classification that we explain in line 67-70.
In the last few years, there have been many classic networks related to Object Classification [6], for instance AlexNet [7] (2012), VGG [8] (2014), GoogLeNet [9] (2015-2016), ResNet [10][11] (2016), SqueezeNet [12] (2016), Xception [13] (2016), MobileNet [14] (2017-2018), ShufficNet [15] (2017-2018), SE-Net [16] (2017), DenseNet [17] (2017), and CondenseNet [18] (2017).
- P-1, L-19:” Our findings show that Yolo V3 SPP strikes the best total BFLOPS (65.69), and mAP (98.88%). Hence, SPP can improve the performance of all models in the experiment.” SPP improves the performance of all models, it is recommended to give specific data to illustrate.
We give specific data to illustrate the result on the line 19-23.
Our findings show that Yolo V3 SPP strikes the best total BFLOPS (65.69), and mAP (98.88%). Besides, the highest average accuracy is Yolo V3 SPP at 99%, followed by Densenet SPP at 87%, Resnet 50 SPP at 70%, and Tiny Yolo V3 SPP at 50%. Hence, SPP can improve the performance of all models in the experiment.
- P-3, L-113: “The fixed-dimensional vectors are the input to the fully-connected layer. By using SPP, the input image can vary in size, which allows not only arbitrary aspect ratios but also enables absolute scales.” Different sizes of images, different sizes of images obtained by the same convolution, and then through the full connection layer fixed by neurons are easy to cause problems.
Thanks for the comments. We apply data pre-processing, so it will no problem if the same convolution obtains different sizes of images. Then, through the full connection layer fixed by neurons, it is easy to cause problems.
- P-3, L-118: “In our work, the SPP blocks layer is inserted to the Yolo V3, Resnet 50, Densenet, and Tiny Yolo V3 configuration file.” Resnet50 and Densenet classification use fully connected layers, while Yolov3 uses convolutional layer prediction. Whether the SPP structure used by three is the same, please explain.
Yes, we use the same SPP structure for Yolo V3, Resnet 50, Densenet, and Tiny Yolo V3 configuration file. We add the explanations in the line 120-125.
In our work, the SPP blocks layer is inserted to the Yolo V3, Resnet 50, Densenet, and Tiny Yolo V3 configuration file. Moreover, we use the same SPP blocks layer in the configuration file with a spatial model. The spatial model uses downsampling in convolutional layers to receive the important features in the max-pooling layers. It applies three different sizes of the max pool for each image by using [route]. Different layers -2, -4 and -1, -3, -5, -6 in were uses in each [route].
- P-3, L-124: “Yolo V3 and Tiny Yolo V3” There are multi-scale predictions for Yolo v3 and yolov3-tiny. In this section, please explain where SPP is added to Yolov3 and yolov3-tiny.
We add the explanations of SPP in the line 159-162.
Further, Yolo V3 SPP and Tiny Yolo V3 SPP is implemented by incorporating three SPP modules in Yolo V3 and Tiny Yolo V3 between the 5 and 6 convolutional layers in front of three detection headers. Yolo V3 SPP and Tiny Yolo V3 SPP are designed to improve the detection accuracy of baseline models further.
- P-7, L-234: “The dataset split into 70% for training and 30% for testing. This experiment focused on Taiwan's prohibitory sign that consists of 235 no entry images,250 no stopping images, 185 speed limit images, and 230 no parking images.” Whether the data set contains pictures of multiple scenes and weather.
Thanks for the useful comment. We do not use traffic sign images with different weather in this experiment, but we add in our future work.
- P-11, L-332: “Table 2. Training loss value, mAP, and AP performance for class P1, P2, P3, and P4.” Whether the GPU is used for acceleration in Table 2. If yes, check if the time is wrong, please check carefully.
Yes, GPU is used for acceleration in Table 2. The detection time meaning is only the calculation time of the best weights to calculate the AP (%), TP, FP, Precision, Recall, F1-score, IoU (%), and mAP@0.50 (%). So, to avoid confusion with the detection time, we delete it.
- P-12, L-354: “Table 3. Testing accuracy using Taiwan’s prohibitory sign images of various sizes.” Testing a single picture is not very persuasive, it is recommended to conduct an overall statistical analysis of the test data set.
Thanks for the comment; we will add more pictures for testing in the future work.
- It is recommended to add experimental data in multiple scenarios and multiple weathers to show that the algorithm is robust.
We have provided the results of our experiments with several scenarios, such as daylight and night light in Figure 9 and Figure 10. Also, we will add experiment data in multiple scenarios and weather in future work. More explanations are in line 428-431.
In future work, we will enhance our dataset to all of Taiwan’s traffic signs and add experimental data in multiple scenarios and different weather conditions for training and testing. We will expand the dataset through Generative Adversarial Networks (GAN) [62][63][64] to obtain better performance and results.
Reviewer 3 Report
This paper analyzes and compares eight Convolutional Neural Network models with Spatial Pyramid Pooling for traffic sign detection.
This is well-written and can be published after some minor modifications.
First, in the References sections, formatting of items needs to be consistent. For example, item 8, item 9 and item 14 are all different.
Second, in equation 13, “-a log a” should be –a log a_hat. Please pay more attention to other equations for correctness, too.
On page 1, line 40: “are easier detection by” may be “are easier to detect by.”
On page 6, line 202: “NMS used” should be “NMS is used.”
On page 8, line 277: What do “scale=0.1, 0.1” and “steps=6400, 7200” mean? Do you mean the former is for YOLO V3 and the latter for YOLO V3 SPP? Please be precise.
Author Response
Thanks for your useful comments. We explain the question as follows.
- First, in the References sections, formatting of items needs to be consistent. For example, item 8, item 9 and item 14 are all different.
We follow the reference format based on the template of “Electronics”. Reference 8 is a conference paper, reference 9 is a journal paper, and reference 14 is a conference paper, that is why the format is different.
- Second, in equation 13, “-a log a” should be –a log a_hat. Please pay more attention to other equations for correctness, too.
Thanks for the comments. We check, revise all equations and ensure that they are correct. (Line 261)
- On page 1, line 40: “are easier detection by” may be “are easier to detect by.”
We revised lines 41-43 based on reviewer’s comments.
These variations are easier to detect by humans, but they may pose a significant challenge for an automatic detection system.
- On page 6, line 202: “NMS used” should be “NMS is used.”
We checked and revised all grammar errors in line 212-213 based on reviewers’ comments.
NMS is used to perform a maximum local search to suppress redundant boxes and output, then display the results of object detection.
- On page 8, line 277: What do “scale=0.1, 0.1” and “steps=6400, 7200” mean? Do you mean the former is for YOLO V3 and the latter for YOLO V3 SPP? Please be precise.
We give explanations in the line 282-293. The difference between Yolo V3 and Yolo V3 SPP is the SPP block. The SPP block inserts into the Yolo V3 configuration file, and we named it Yolo V3 SPP. Yolo V3 and Yolo V3 SPP using the same configuration setting such as max_batches = 8000 iterations, policy=steps, scale = 0.1,0.1 and steps = 6400, 7200.
Figure 5 explains the reliability of the training process using Yolo V3 (a) and Yolo V3 SPP (b). The training loss value for each model is 0.0141 and 0.0125, respectively. Our work uses max_batches = 8000 iterations, policy=steps, scale = 0.1,0.1 and steps = 6400, 7200. At the beginning of the training process, the system is starting with zero information and a high learning rate. Therefore, as the neural network is presented with growing amounts of data, the weights must change less aggressively. Thus, the learning rate needs to be decreased over time. Further, in the configuration file, this decrease in learning rate is accomplished by first specifying that our learning rate decreasing policy is stepwise. For instance, the learning rate starts from 0.001 and remains constant for 6400 iterations. It then multiplies by scales to obtain the new learning rate. If the scale = 0.1, 0.1 and the current iteration number is 10000 (0.001) then current_learning_rate = learning_rate * scales [0] * scales [1] = 0.001 * 0.1 * 0.1 = 0.00001. From Figure 5, we can conclude that Yolo V3 SPP is more stable than Yolo V3 during the training process.